# Ambient hydrogenation of solid aromatics enabled by a high entropy alloy nanocatalyst

Zekun Jing[1,7], Yakun Guo[1,7], Qi Wang [1,7], Xinrong Yan[2], Guozong Yue[3], Zhendong Li[1], Hanwen Liu[4], Ruixuan Qin [5], Changyin Zhong[3], Mingzhen Li[1,6], Dingguo Xu [2] ✉, Yunxi Yao [3] ✉, Yonggang Yao [4] ✉ & Maobing Shuai[3] ✉

Hydrogenation is a versatile chemical process with significant applications in various industries, including food production, petrochemical refining, pharmaceuticals, and hydrogen carriers/safety. Traditional hydrogenation of aromatics, hindered by the stable π-conjugated phenyl ring structures, typically requires high temperatures and pressures, making ambient hydrogenation a grand challenge. Herein, we introduce a PdPtRuCuNi high entropy alloy (HEA) nanocatalyst, achieving an exceptional 100% hydrogenation of carbon-carbon unsaturated bonds, including alkynyl and phenyl groups, in solid 1,4-bis(phenylethynyl)benzene (DEB) at 25 °C under ≤1 bar $H_2$ and solventless condition. This results in a threefold higher hydrogen uptake for DEB-contained composites compared to conventional Pd catalysts, which can only hydrogenate the alkynyl groups with a ~ 27% conversion of DEB. Our experimental results, complemented by theoretical calculations, reveal that PdPtRu alloy is highly active and crucial in enabling the hydrogenation of phenyl groups, while all five elements work synergistically to regulate the reaction rate. Remarkably, this newly developed catalyst also achieves nearly 100% reactivity for ambient hydrogenation of a broad range of aromatics, suggesting its universal effectiveness. Our research uncovers a novel material platform and catalyst design principle for efficient and general hydrogenation. The multi-element synergy in HEA also promises unique catalytic behaviors beyond hydrogenation applications.

The catalytic hydrogenation reaction has significant applications in the energy, chemistry, and pharmaceutical industries[1–3]. For example, chemical products such as alkanes, alkenes, alcohols, and amines are usually synthesized by hydrogenating −C = C−, −C ≡ C−, −C = O, −NO₂, and −COOH functional groups containing unsaturated bonds[4–8]. Particularly, the hydrogenation of carbon-carbon (C-C) unsaturated bonds is one of the most intensively investigated topics for synthesizing raw materials in the fine chemical industry, preparing liquid organic hydrogen carriers, and ensuring hydrogen safety applications[5,6,9–13]. However, compared to the easy hydrogenation of ene/alkyne groups, hydrogenating π-conjugation phenyl group (i.e., aromatics) presents a notable challenge, due to their high aromaticity,

[1]Science and Technology on Surface Physics and Chemistry Laboratory, Jiangyou 621908, China. [2]College of Chemistry, Sichuan University, Chengdu 610065, China. [3]Institute of Materials, China Academy of Engineering Physics, Mianyang 621907, China. [4]State Key Laboratory of Materials Processing and Die & Mould Technology, School of Materials Science and Engineering, Huazhong University of Science and Technology, Wuhan 430074, China. [5]Department of Chemistry, College of Chemistry and Chemical Engineering, Xiamen University, Xiamen 361005, China. [6]State Key Laboratory of Environment-friendly Energy Materials, Southwest University of Science and Technology, Mianyang 621010, China. [7]These authors contributed equally: Zekun Jing, Yakun Guo, Qi Wang. ✉e-mail: dgxu@scu.edu.cn; yaoyunxi@caep.cn; yaoyg@hust.edu.cn; shuaimb@sina.com

nonpolarity, and large reaction energy barriers involved[14–16]. Typically, hydrogenating aromatic rings requires higher reaction temperature (>100 °C) and hydrogen pressure (up to 50 bar), even under the use of noble catalysts[12,17,18]. Although certain compounds (e.g., benzene) can be hydrogenated at mild conditions[19–21], solvents (e.g., isopropanol or ethanol)[19,20] or promoters (e.g., Lewis acid)[21] are required to facilitate the hydrogen transfer and ring open process in the aromatics.

Ambient hydrogenation is not only facile and energy efficient (no need for energy input) but also essential for some hydrogen-related applications. For instance, solid hydrogen getters are required to control the hydrogen concentration and reduce the risk of explosion in hydrogen stations and cells. In some battery-driven equipment, hydrogen is inevitably released, which brings a severe explosion risk for defense and energy applications. Hence, solid organic hydrogen getters are indispensable to effectively hydrogenate at room temperature without any solvents or promoters due to equipment constraints and environmental limitations. As a typical example, the 1,4-bis(phenylethynyl)benzene (DEB) molecules, linked by the alkyne and phenyl ring groups, are commonly used as hydrogen getters[13,22–24]. However, catalyzed by palladium catalyst, only the alkynyl groups are hydrogenated at ambient temperature. To the best of our knowledge, the general and complete hydrogenation of solid aromatics under ambient, solvent-free conditions has yet to be realized, representing a grand challenge both in scientific realization and practical applications.

Theoretically, the hydrogenation of solid aromatics requires highly efficient catalysts capable of simultaneously activating $H_2$ and C-C unsaturated bonds. However, most existing catalysts, limited to one or a few metals, offer inadequate active sites and are incapable of realizing full hydrogenation, especially for robust aromatics. Recently, high-entropy alloy (HEA) nanocatalysts have been widely used in catalysis due to their flexible composition, high activity, excellent stability, and a magic cock-tailing effect[25–29], which offer the promise of catalytic properties that could exceed conventional catalysts[26,30–32].

HEA catalysts, containing multifunctional active sites, are ideal for stabilizing various intermediates, optimizing adsorption energy, and facilitating multi-step reactions such as hydrogenation[29].

Here, we report the design and synthesis of PdPtRuCuNi HEA nanocatalysts, supported on the $HNO_3$-modified carbon nanofibers (CNFs) via a simple solvothermal method. These catalysts achieve complete hydrogenation of C-C unsaturated bonds in solid DEB (~100% conversion), including both alkynyl and phenyl groups, in a solventless manner at 25 °C under ≤1 bar $H_2$ (Fig. 1a). Compared with traditional Pd/C catalysts, which only activate alkynyl groups, this results in a > 3 times higher hydrogen uptake capacity and conversion. Our HEA catalyst design incorporates Pd as a common element for hydrogenating alkynyl groups[5,9,33–36], while Pt, Ru, Ni, etc., are suitable for activating phenyl groups[14,21,22,37,38]. Experimental and theoretical analyses indicate that PdPtRu alloying is critical for hydrogenating phenyl groups, with all elements collectively regulating the hydrogenation reaction rate. This catalyst strategy also proves effective for hydrogenating most aromatics at room temperature with nearly 100% reactivity (Fig. 1b). Our findings thus provide a novel material platform and catalyst design guideline for highly active, cost-effective, and safe hydrogenation of benzene and its derivatives.

## Results

### Synthesis and characterization of PdPtRuCuNi/CNFs

The PdPtRuCuNi HEA nanoparticles were synthesized with acetylacetone salts as metal precursors (Supplementary Fig. 1)[39]. The atomic ratio of Pd, Pt, Ru, Cu, and Ni is nearly 21:13:20:24:22 (summarized in Supplementary Table 1), as measured by inductively coupled plasma optical emission spectroscopy (ICP-OES). The HEA particles are supported uniformly on CNFs and have an average size ($d_{ave}$) of approximately $17.5 \pm 12.5$ nm (Fig. 2a, b and supplementary Fig. 2a). The high-resolution transmission electron microscopy (HRTEM) suggests that the PdPtRuCuNi HEA particles are nanocrystalline with an interplanar spacing of about 2.20 Å (Fig. 2c and sup-

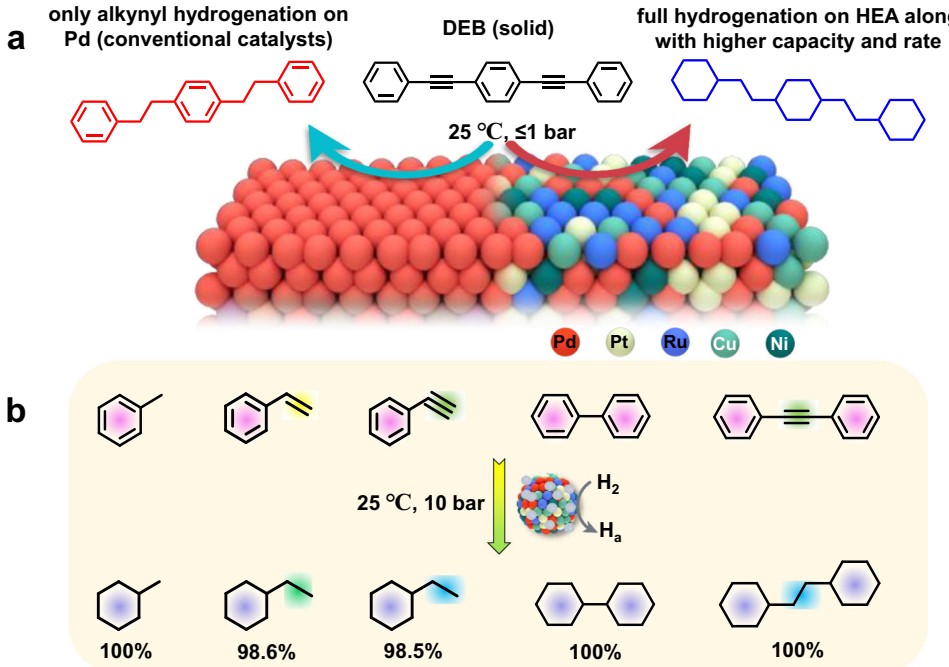

**Fig. 1 | Hydrogenation of aromatics by PdPtRuCuNi HEA catalyst at room temperature. a** The hydrogenation schematic of solid DEB molecules catalyzed by Pd (left) and PdPtRuCuNi HEA (right) catalysts at 25 °C under ≤1 bar $H_2$. **b** The conversion of different liquid and solid aromatics at 25 °C under 10 bar $H_2$ in a solvent-free state. Liquid reaction condition: 40 mg PdPtRuCuNi HEA/CNFs, 3 mL toluene, styrene, or phenylacetylene, 10 h. Solid reaction condition: 10 mg PdPtRuCuNi HEA/CNFs, 30 mg biphenyl or diphenylacetylene, 36 h.

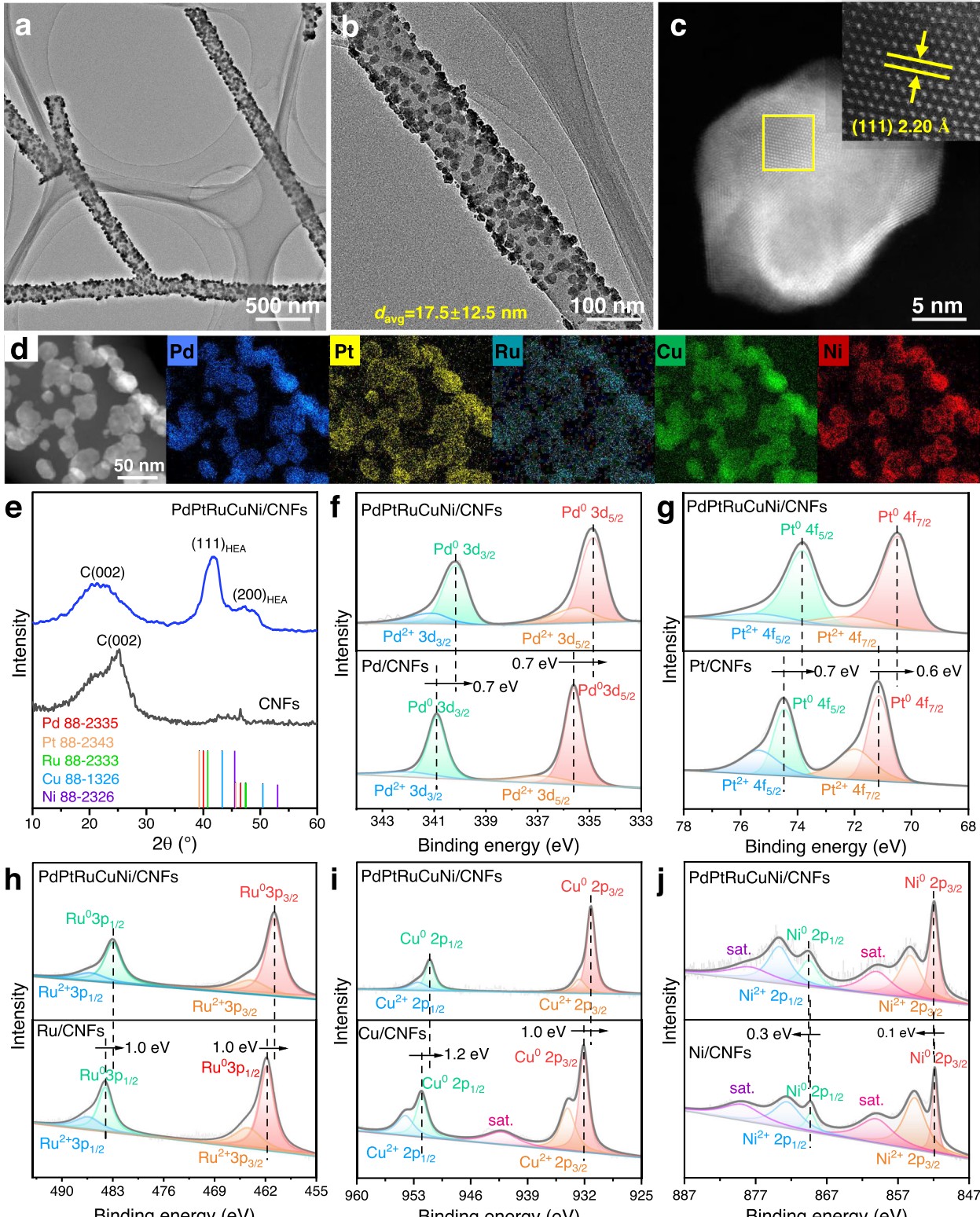

**Fig. 2 | Characterization of PdPtRuCuNi/CNFs. a, b** The morphologies. **c** HRTEM images. **d** EDS mapping. **e** XRD pattern. **f-j** The high-resolution XPS spectra of PdPtRuCuNi HEA and monometal catalyst. **f** Pd 3*d*, **g** Pt 4*f*, **h** Ru 3*p*, **i** Cu 2*p*, and **j** Ni 2*p*.

plementary Fig. 2b). High-angle annular dark-field scanning transmission electron microscopy (HAADF-STEM) and energy-dispersive X-ray spectroscopy (EDS) mapping reveal that all elements are homogeneously distributed within the PdPtRuCuNi HEA nanoparticles (Fig. 2d and Supplementary Fig. 2c). The PdPtRuCuNi/CNFs exhibit two broad X-ray diffraction (XRD) peaks at 2θ of 41.4 ° and 47.4 °, corresponding to the (111) and (200) crystalline planes of PdPtRuCuNi HEA (Fig. 2e and Supplementary Table 3), indicating a single-phase alloy with a fcc structure. X-ray photoelectron spectroscopy (XPS) analysis shows that the metals exist mainly in their zero-valence states (Fig. 2f-

j). A small number of metal oxides are present, likely originating from surface oxidation. Compared to their individual metals, the binding energies of Pd 3$d$, Pt 4$f$, Ru 3$p$, and Cu 2$p$ shift to the lower energies (Fig. 2f–i), while Ni 2$p$ exhibits a slight shift to higher energies (Fig. 2j). This suggests that alloying these metals result in a strong electronic interaction within the PdPtRuCuNi HEA, which could change the surface electronic structure and potentially enhance its catalytic performance[40]. These results have also been demonstrated by X-ray absorption near edge structure (XANES) spectra (Supplementary Fig. 4a-c). The Pt, Cu, and Ni absorption energies for PdPtRuCuNi/CNFs are similar to those of metal foil, indicating the dominance of metallic-stated elements. The last half is slightly changed in shape and intensity, indicating the interaction among each element. The Fourier transforms of these EXAFS are displayed in Supplementary Fig. 4d-f. The distances of Me−Me in PdPtRuCuNi HEA are overall shorter than that of metal foils, and the line intensity of PdPtRuCuNi HEA is much lower than that of the metal foil, indicating the change in bond lengths and coordination numbers. The bond length (R) and coordination numbers are summarized in Supplementary Table 4.

## Catalytic hydrogenation of alkynyl and phenyl groups

The solid DEB molecule was selected as a representative reactant to evaluate the catalytic activity of PdPtRuCuNi/CNFs. They were mixed uniformly with PdPtRuCuNi/CNFs and Pd/CNFs to form loose composites, providing more accessible pathways for H$_2$ diffusion (Supplementary Fig. 5 and 6). The hydrogen uptake tests were performed at different temperatures under ≤1 bar H$_2$, recording the time and uptake capacity automatically. Both the PdPtRuCuNi/CNFs-DEB and Pd/CNFs-DEB composites exhibit increasing hydrogen uptake with reaction time at 25 °C (Fig. 3d and Supplementary Fig. 7a). Notably, the final hydrogen uptake capacities for PdPtRuCuNi/CNFs-DEB and Pd/CNFs

composites are 762 cm$^3$/g and 198 cm$^3$/g, significantly surpassing those of the catalysts themselves without DEB ($\sim$2 cm$^3$/g for PdPtRuCuNi/CNFs, and $\sim$30 cm$^3$/g for Pd/CNFs, Fig. 3a). This indicates that the hydrogen uptake is primarily attributed to the chemisorption (hydrogenation) of DEB molecules, affecting significantly by the element type. In addition, the commercial Pd/C and Ru/C are used as contrast catalysts, the hydrogen uptakes of Pd/C-DEB and Ru/C-DEB composites are 186 cm$^3$/g and 0 cm$^3$/g (Fig. 3b and supplementary Fig. 7), respectively, which are significantly lower than PdPtRuCuNi/CNFs, and Ru/C with no catalytic activity for DEB in mild condition.

We use the slopes of the hydrogen uptake curve to denote the reaction rate of DEB, and the linearly fitted rates of different stages are marked in Fig. 3c, d. Firstly, the hydrogen uptake curves of PdPtRuCuNi/CNFs-DEB were evaluated at various temperatures (25, 50, and 100 °C, Fig. 3c and supplementary Fig. 8). The results show that the similar hydrogen uptake values are obtained at 25 and 50 °C, while some decrease of hydrogen uptake at 10 °C because increasing the temperature promote the chemical reaction equilibrium moving in the opposite direction due to the exothermic hydrogenation reaction. Moreover, the reaction rates are elevated from 7.12 cm$^3$/(g·h) to 25.6 cm$^3$/(g·h) as the temperature from 25 °C to 100 °C in the initial hydrogenation process. However, the reaction rate falls sharply as the times increase for 50 °C and 100 °C, and the curves have a significant inflection point, which means the reaction orders and kinetics have changed. So PdPtRuCuNi has a high activity at 25 °C. Next, the Pd/CNFs-DEB as the control group to further elevate the reaction rate, Pd/CNFs-DEB reacts rapidly with H$_2$, with a rate of about 12.85 cm$^3$/(g·h), and then the reaction rate decays with time. In contrast, PdPtRuCuNi/CNFs-DEB displays a nearly constant reaction rate of $\sim$7.12 cm$^3$/(g·h) until reaching its maximum hydrogen uptake. This suggests that the PdPtRuCuNi HEA catalyst demonstrates excellent catalytic

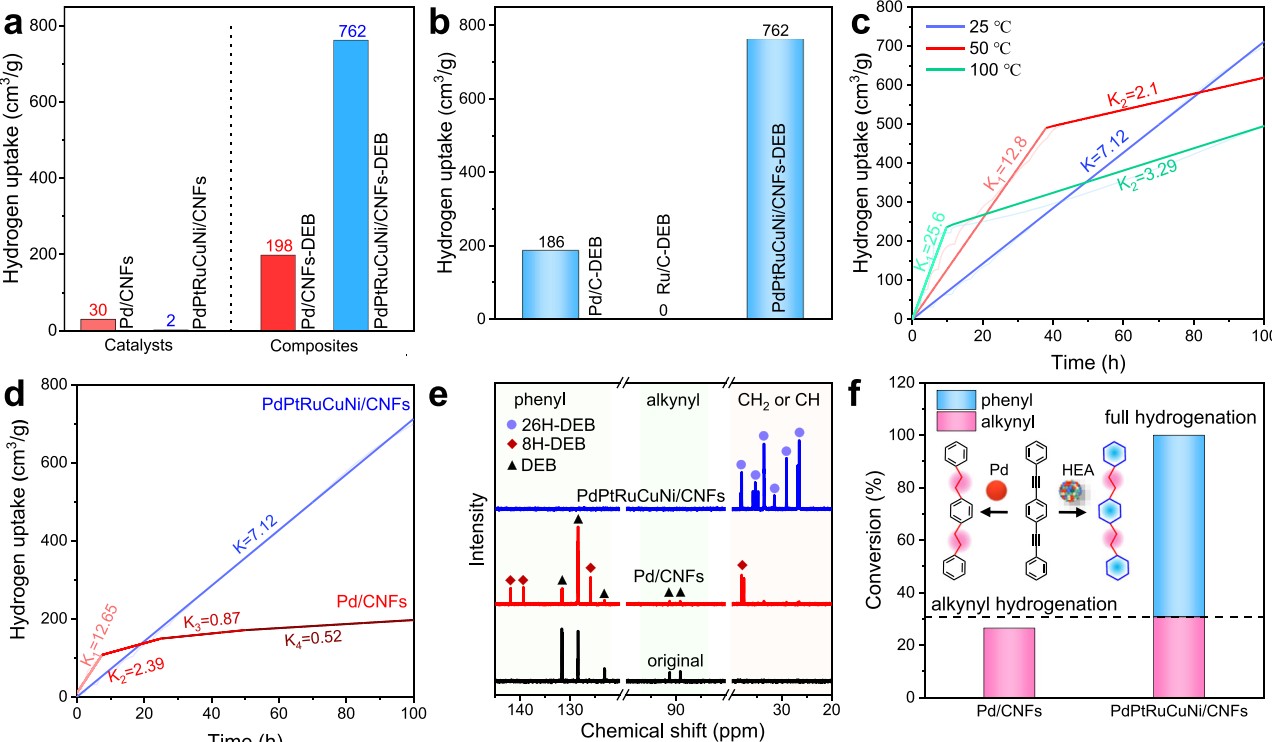

**Fig. 3 | The hydrogenation performances of DEB catalyzed by PdPtRuCuNi/CNFs and Pd/CNFs at 25 °C under ≤ 1 bar H$_2$. a** The final hydrogen uptake capacities of PdPtRuCuNi/CNFs-DEB and Pd/CNFs-DEB composites compared with catalysts themselves. **b** The final hydrogen uptake capacities of commercial Pd/C-DEB, Ru/C-DEB compared with PdPtRuCuNi/CNFs-DEB. **c** The hydrogen uptake curves with time at different temperature. **d** The curves of hydrogen uptake and reaction time of PdPtRuCuNi/CNFs-DEB and Pd/CNFs-DEB. **e** $^{13}$C NMR spectra of DEB before and after hydrogenation. **f** The conversion of C-C unsaturated bonds (alkynyl and phenyl groups) in the hydrogenated DEB molecules.

performance in DEB hydrogenation and operates through a different catalytic mechanism with Pd/CNFs. This difference is likely caused by the synergistic effects of multiple elements in HEA during the solventless hydrogenation process.

The hydrogenated products are analyzed using NMR spectra. As shown in Fig. 3e, f and Supplementary Fig. 7c, the Pd/CNFs catalyst only hydrogenates the alkynyl group in the DEB molecules (88% conversion of all the alkynyl groups), forming 1,4-bis(phenylethyl)benzene (8H-DEB). For the PdPtRuCuNi/CNFs, they display outstanding catalytic activity towards both alkynyl and phenyl groups, achieving complete hydrogenation and forming 1,4-bis(ethylcyclohexane) cyclohexane (26H-DEB). This transformation brings almost 100% conversion of the C-C unsaturated bonds in DEB catalyzed by PdPtRuCuNi/CNFs, while achieving ~27% conversion by Pd/CNFs, approximately three times increased, which agrees well with the hydrogen uptake capacities. Moreover, the PdPtRuCuNi/CNFs-DEB composite evolves from powders into melts after hydrogenation, indirectly indicating the deep hydrogenation of DEB molecules (Supplementary Fig. 6). Overall, the utilization of a novel PdPtRuCuNi HEA nanocatalyst has led to a breakthrough in achieving the simultaneous and complete hydrogenation of alkynyl and phenyl groups under ≤1 bar hydrogen pressure, marking a notable advancement in this field.

## Synergistic catalytic effect of PdPtRuCuNi

To delve deeper into the synergistic effect of the PdPtRuCuNi catalyst and clarify the roles of each element in the hydrogenation of DEB, we have synthesized a series of quaternary nanocatalysts with similar loadings (supplementary table 5) by removing one element at a time (i.e., PtRuCuNi/CNFs, PdRuCuNi/CNFs, PdPtCuNi/CNFs, PdPtRuNi/CNFs, and PdPtRuCu/CNFs), and their phase structures, chemical compositions, and morphologies have been characterized in detail (Supplementary Fig. 9-19). The catalytic performances of these catalysts for the alkynyl and phenyl groups in the DEB molecules have been investigated and the reaction rates and hydrogenated products are analyzed following the same procedures stated above (Fig. 4 and Supplementary Fig. 20). The reduction in reaction rate occurs in descending order as the Pd, Pt, Ru, Cu, and Ni are sequentially removed, with the largest decrease occurring when Pd is removed and the smallest when Ni is removed (Fig. 4a and Supplementary Fig. 20a). It is noteworthy that the reaction rates of these quaternary catalysts are all lower than that of the PdPtRuCuNi, indicating that all five elements work synergetically to regulate the reaction kinetics of DEB. Importantly, when any one of the Pd, Pt, or Ru elements are absent, the composites display not only decreased reaction rates but also a reduction in final hydrogen uptake capacity and conversion of DEB (Fig. 4b and Supplementary Fig. 20b). In contrast, when these three essential elements−Pd, Pt, and Ru−coexist, the composites exhibit robust hydrogen uptake performance, achieving nearly 100% DEB conversion. These findings have also been demonstrated by PdPtRu/CNFs catalyst (Supplementary Fig. 21 and 22), indicating that the Pd, Pt, and Ru elements play a vital role in the hydrogenation of DEB.

A detailed comparison from NMR analysis shows that the difference in the total conversion of these quaternary catalysts mainly stems from the hydrogenation of phenyl groups (Fig. 4c and Supplementary Fig. 20c). Specifically, under the catalysis of PtRuCuNi and PdRuCuNi, the primary products are 8H-DEB, with only alkynyl groups hydrogenated. When using PdPtCuNi as the catalyst, the products transform into a mixture of 1-ethylcyclohexane-4-phenylethyl benzene and 1,4-bis(ethylcyclohexane)benzene, termed 14H- and 20H-DEB, with one or two sides of phenyl groups further hydrogenation. When Pd, Pt, and Ru coexist, the product becomes 26H-DEB, with alkynyl and phenyl groups undergoing complete hydrogenation (Supplementary Fig. 20d). Thus, we deduce that alkynyl groups are most readily hydrogenated, then one or two sides of phenyl groups, and finally the middle phenyl group. This hierarchy in the hydrogenation process of DEB can be attributed to the different activation energies associated with phenyl groups at different sites.

## Understanding the hydrogenation process over PdPtRuCuNi HEA

We proceed to gain insights into the synergistic effect of elements within the PdPtRuCuNi HEA in the context of hydrogenation. As the activation of both $H_2$ and aromatic molecules are critical for hydrogenation, we utilize density functional theory (DFT) calculations to unveil their interactions with the close-packed (111) surfaces of PdPtRuCuNi HEA and Pd, Pt, Ru, Cu, Ni single metals. To save computation costs, we employ phenylacetylene ($C_6H_5-C\equiv CH$), possessing phenyl and alkynyl groups, as a model molecule of DEB (abbreviated as PhA hereafter). In the later section, we also verify that the conversion of PhA is ~98.5%, close to 100%. As to the favored adsorption sites, we consider both hollow and bridge sites for $H_2$ and H species ($H_a$). For PhA, previous studies, verified by our tests, indicate that the optimal adsorption configuration involves the phenyl group sitting above a 3-fold hollow site ($\pi_{ph}$), with the alkynyl group forming di-$\mu$ bonding[41]. Furthermore, we've constructed six random slabs for the PdPtRuCuNi HEA and explored all possible adsorption sites on the slabs to evaluate the impact of local atomic arrangement on its catalytic activity (Methods).

We start with the adsorption of $H_2$ molecules. The adsorption energies (denoted as $\Delta E_{ads-H2}$) and the optimized H-H distance for 480 adsorption sites (192 hollow and 288 bridge sites, see Methods) on the PdPtRuCuNi slabs are shown in Supplementary Fig. 26. All $H_2$ molecules are activated on the PdPtRuCuNi surface, with H-H bonds elongated ($d_{H-H} > 0.74$ Å). At ~47.5% of the adsorption sites, the $H_2$ molecule dissociates, with a final H-H distance > 1.48 Å (twice the equilibrium H-H bond length) and $H_a$ as the product. This dissociation also pulls the $H_a$ closer to the HEA surface. In comparison, the $H_2$ dissociates completely on the Pd catalyst's surface (Supplementary Table 6). Thus, we infer that the slight reduction in the reaction rate of PdPtRuCuNi/CNFs-DEB compared with Pd/CNFs-DEB at the initial stage in Fig. 3a might be attributed to an insufficient $H_a$ supply under low $H_2$ pressure. Supplementary Fig. 27 illustrates the spatial variation of $\Delta E_{ads-H2}$ on the six random slabs, where site-to-site heterogeneity is observed, suggesting a large variety on a single HEA surface. We selected an active site near Pd from Slab 6 and derived the $H_2$ dissociation energy via the climbing image nudged elastic band method (CI-NEB)[42]. This site displays the lowest $H_2$ dissociation energy compared to any other single metal, despite all the metals except Cu having relatively low dissociation energies (<0.1 eV) (Supplementary Fig. 28a).

The migration of $H_a$ from the catalyst to the reactant is crucial for hydrogenation, significantly affecting the reaction rate and ultimate hydrogenation conversion. We calculate the binding energies of the dissociated $H_a$ with the adsorption sites on Slab 6 of PdPtRuCuNi (Fig. 5a). Compared with single metals, the PdPtRuCuNi HEA display a broader distribution of binding energies. About 86.5% and 40.6% of the sites possess lower binding energies than those on the hollow and bridge sites of the pure Pd surface, due to the weak binding of Pt and Cu with $H_a$. It indicates that most $H_a$ bind weakly with PdPtRuCuNi and are easy to transfer from the catalyst to the reactant. Our temperature-programmed desorption (TPD) data validates these DFT calculation results, as evidenced by the broader desorption peak of $H_2$ and a ~7.8 °C decrease in the maximum desorption peak of PdPtRuCuNi/CNFs compared to that of Pd/CNFs (Supplementary Fig. 28b).

Next, we examine the interaction of PhA with the HEA surface. Figure 5b displays the adsorption energy distribution of PhA ($\Delta E_{ads-PhA}$) on the HEA and monometallic slabs. In the monometallic case, the $\Delta E_{ads-PhA}$ on Pd, Pt, Ru, Cu, and Ni surfaces are −3.13, −2.66, −4.99, −0.88, and −3.78 eV, respectively. After alloying into PdPtRuCuNi, the $\Delta E_{ads-PhA}$ exhibits a widened distribution from −1.47 to −4.41 eV,

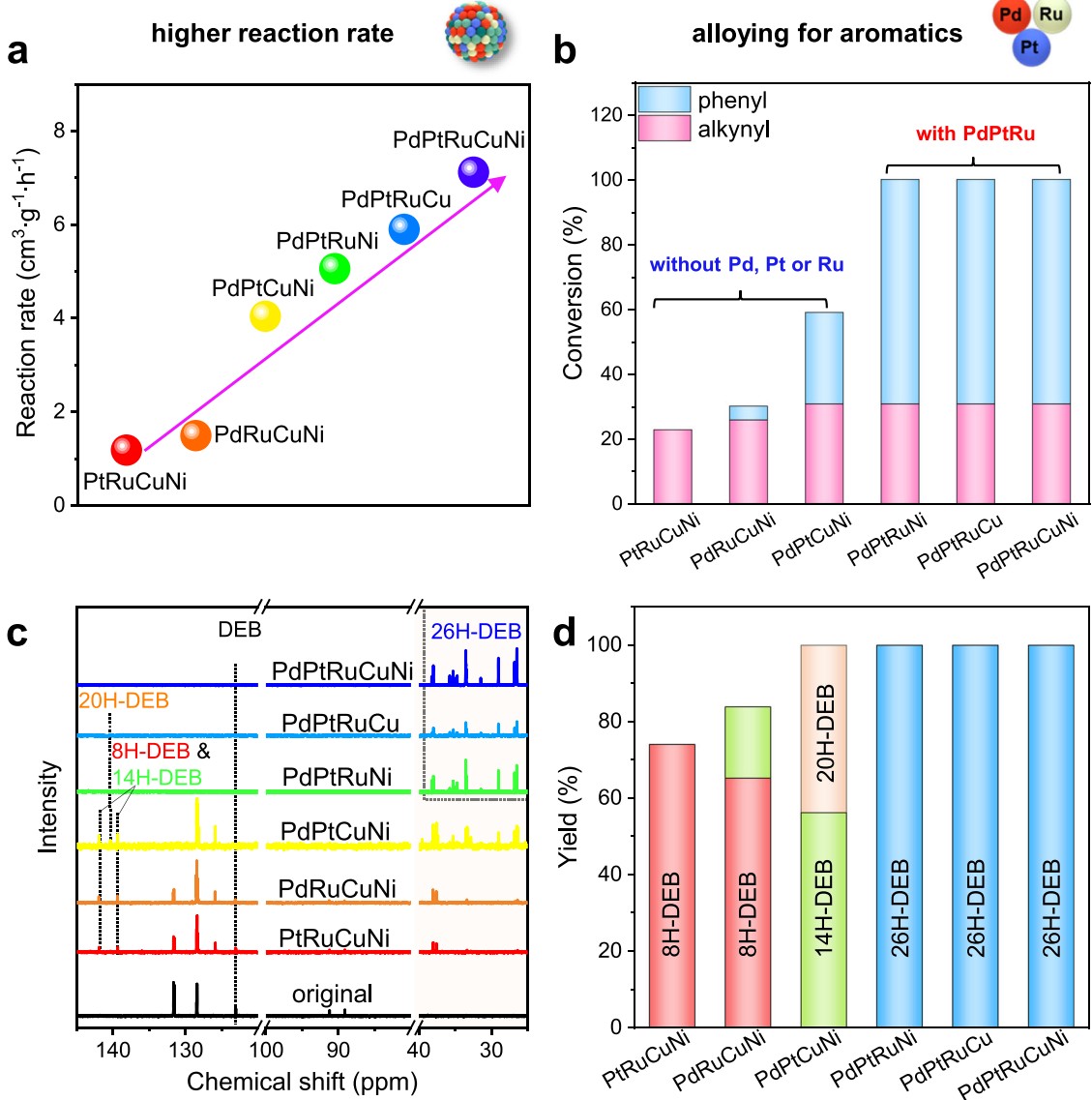

**Fig. 4 | The synergistic catalytic effect of PdPtRuCuNi HEA. a** The reaction rates of the DEB-contained composites catalyzed by quaternary and quinary catalysts. **b** The overall conversion of C-C unsaturated bonds in DEB and the relative conversion of alkynyl (pink) and phenyl (blue) groups. **c** $^{13}$C NMR spectra of DEB before and after hydrogenation. **d** The yields of different hydrogenated DEB products.

reminiscent of a Gaussian shape with its peak near the Pd catalyst. Supplementary Fig. 29 gives the spatial distribution of $\Delta E_{ads\text{-}PhA}$ on the slabs as in Supplementary Fig. 27. This diversity can effectively regulate the adsorption energies (or behaviors) of PhA and its intermediates, creating an undulating landscape for them to adsorb and desorb[29]. Supplementary Fig. 30a presents the increase in C-C bond distances within the alkynyl and phenyl groups after activation by Pd and PdPtRuCuNi HEA catalysts. We see the C-C distances in the two groups increase, with the elongation within the alkynyl group being more pronounced. The alkynyl group is also dragged closer to the PdPtRu-CuNi HEA surface compared to the phenyl group (Supplementary Fig. 30b). It is reasonable to consider that the alkynyl group is more prone to hydrogenation than the phenyl group. To validate this conjecture, we calculated the first-step hydrogenation reaction barriers for the alkynyl and phenyl groups in PhA over the PdPtRuCuNi surface (Fig. 5c). Only 0.13 eV is needed for the alkynyl group to add the first $H_a$, while the phenyl group demands 0.43 eV, corroborating the experimental observations in Fig. 3d and Fig. 4b. Consequently, the phenyl hydrogenation is a thorny problem for achieving complete

hydrogenation. Since the alkynyl group is more prone to hydrogenation and all catalysts can hydrogenate the alkynyl group, we proceed to utilize the product after alkynyl group hydrogenation, namely phenylethane ($C_6H_5$–$CH_2$-$CH_3$), as a model molecule to investigate the subsequent hydrogenation of the phenyl group. Similarly, its first-step hydrogenation reaction barriers on PdPtRuCuNi, Pd, and Ru surfaces are calculated (Fig. 5d). The carbon in the para-position (Supplementary Fig. 31) has the lowest hydrogenation reaction barrier. We find that the first-step hydrogenation reaction barriers of phenylethane on the Pd and Ru are 0.58 and 0.56 eV, while on PdPtRuCuNi, 0.35 eV. These findings further indicate that PdPtRuCuNi can be more favorable for hydrogenating the phenyl group than the Pd and Ru catalysts, explaining why PdPtRuCuNi can achieve complete hydrogenation of DEB.

Based on the above experiment and calculation results, we propose the following reaction mechanism (Fig. 5e): Initially, $H_2$ is activated by the PdPtRuCuNi/CNFs HEA catalyst, dissociating into active $H_a$. Due to the low binding energy and undulating energy landscape on the surfaces, $H_a$ can migrate to other sites through

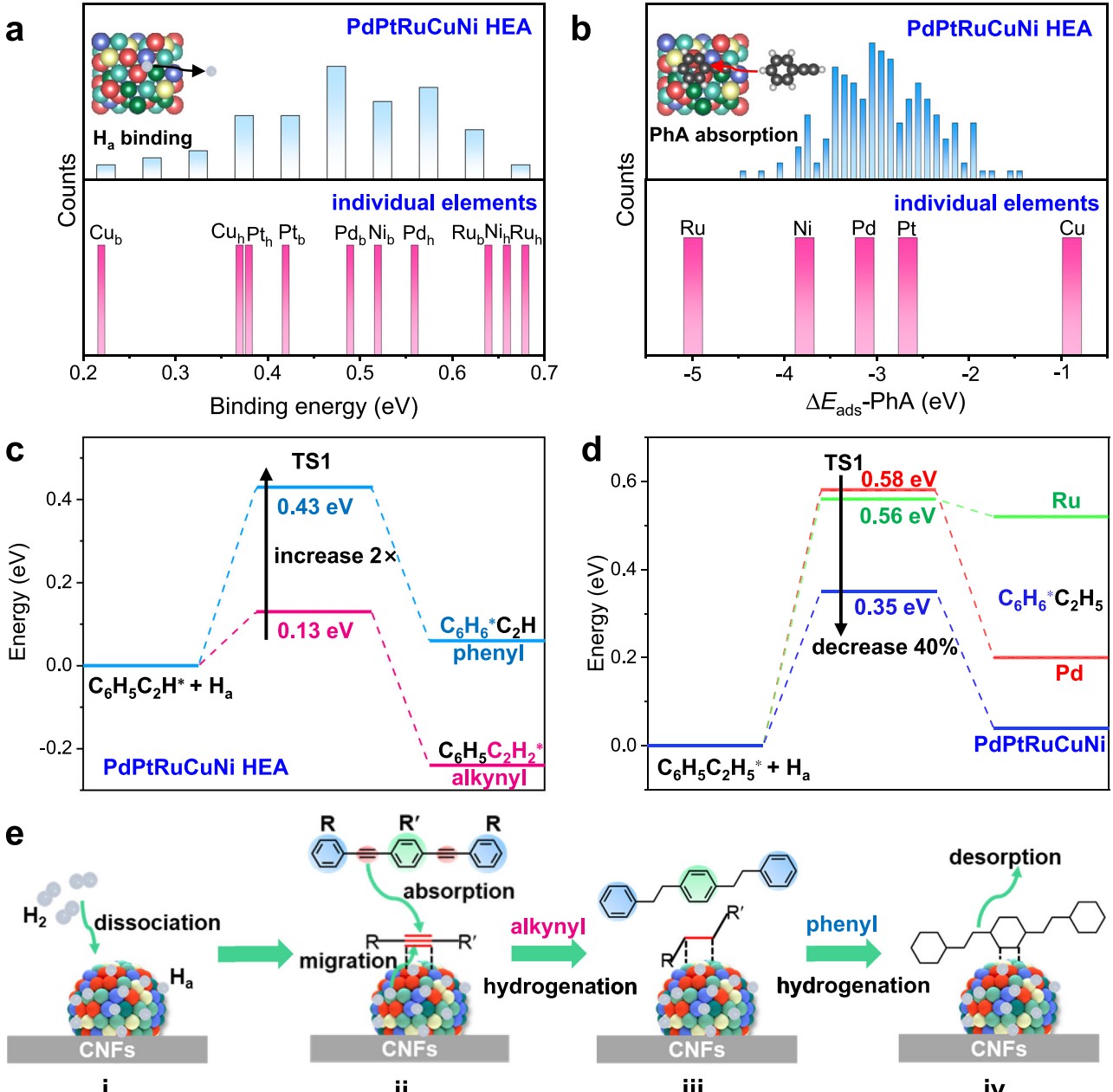

**Fig. 5 | DFT calculation of H₂ and phenylacetylene over the PdPtRuCuNi (111) surface. a** Binding energies of dissociated H species at the hollow and bridge sites of a representative PdPtRuCuNi slab (upper panel) and at the hollow (with subscript "h") and bridge (with subscript "b") sites of Pd, Pt, Ru, Cu, Ni slabs (lower panel). **b** The absorption energy of PhA over the (111) surface of PdPtRuCuNi (above) and Pd, Pt, Ru, Cu, Ni (below) catalysts. **c** The first-step hydrogenation reaction barriers of the alkynyl and phenyl groups in PhA over PdPtRuCuNi (111). **d** The first-step hydrogenation reaction barriers of the phenyl groups in phenylethane over PdPtRuCuNi (111), Pd (111), and Ru (111). **e** Reaction pathways for the hydrogenation of DEB over PdPtRuCuNi HEA nanocatalysts.

diffusion or hydrogen spillover (i). Meanwhile, the DEB molecules are adsorbed and activated on the PdPtRuCuNi HEA, especially around the active sites containing Pd, Pt, and Ru atoms (ii). Then, the $H_a$ meets the activated DEB molecules, and the alkynyl groups are preferentially hydrogenated (iii). Next, one or two sides of phenyl groups in DEB undergo hydrogenation, followed by the middle one, until all phenyl groups are hydrogenated (iv). Finally, the hydrogenated products detach from the PdPtRuCuNi HEA surface. In addition to the reduced reaction barriers for hydrogenation, the existence of diverse and multi-functional sites within the HEA surface enables that the H₂ activation, reactant absorption, and hydrogenation steps can occur separately at different sites, making the hydrogenation more efficient.

## Universal hydrogenation of diverse aromatic derivatives

To assess the versatility of our PdPtRuCuNi HEA catalyst in the hydrogenation of aromatics, we conduct experiments with different aromatic substrates containing phenyl groups. These investigations were conducted at 25 °C in a solvent-free state, with 10 bar H₂ pressure to speed up the reaction. The resulting products were analyzed by NMR, and the conversion was derived (Supplementary Figs. 32–36). Remarkably, nearly 100% reactivity is achieved for toluene, biphenyl, and diphenylacetylene, 98.6% for styrene, and 98.5% for PhA (Fig. 1b). All C-C unsaturated bonds, including ene/alkyne and phenyl groups in different liquid or solid reactants, can be fully hydrogenated at ambient temperature. Notably, these hydrogenations occur with no additional solvents, promoters, or energy inputs. The final hydrogenated

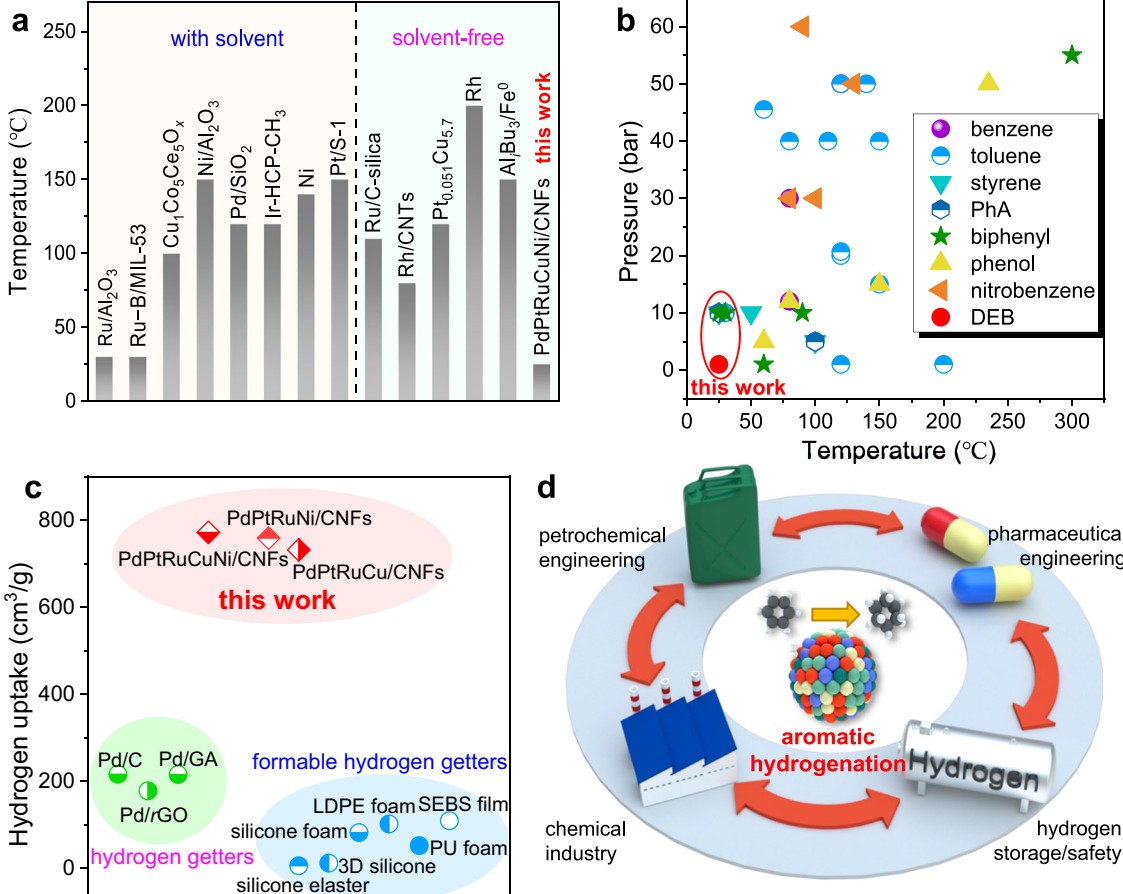

**Fig. 6 | Catalytic hydrogenation of aromatic substrates over PdPtRuCuNi HEA catalyst at 25 °C. a** Catalytic hydrogenation of toluene with different catalysts with or without solvent. **b** The comparison of the hydrogenation conditions of the present study with the reported results in the literature (Supplementary Table 7). **c** The comparison of hydrogen uptake capacities of DEB-contained composites catalyzed by Pd and PdPtRuCuNi HEA catalysts (Supplementary Table 8). **d** The extensive application of aromatic derivatives hydrogenated at ambient conditions.

products are saturated cyclanes, regardless of the presence of substituted groups around the benzene ring (alkyl/ene/alkyne or phenyl), the location of phenyl groups (terminal or internal), or whether the catalysis occurs in a liquid or solid phase. This performance significantly expands the applications of HEA nanocatalysts in the catalytic hydrogenation of various aromatic derivatives.

To highlight the performance of our PdPtRuCuNi HEA catalyst in the hydrogenation of aromatics, we first use toluene as an example and summarize the hydrogenation conditions of toluene catalyzed by different catalysts in Fig. 6a. Compared with Pd, Pt, Ru, and Rh-contained catalysts[17–21,43–46], we achieve a complete conversion and a 100% yield of methylcyclohexane under mild conditions (25 °C, 10 bar $H_2$), without any solvent or promoter. Although similar reaction conditions (30 °C, 10 bar $H_2$) have been reported for benzene hydrogenation over Ru/α-$Al_2O_3$ or Ru-B/MIL-53 (B = Al, Cr, and AlCr)[43,45], these reactions require alcohol solvents such as isopropanol or ethanol (Supplementary Table 7). Moreover, after being used four times, our PdPtRuCuNi HEA catalyst still maintains 100% conversion for toluene, underscoring its stability (Supplementary Fig. 32c and d). Figure 6b and Supplementary Table 7 further give the temperature-pressure conditions for hydrogenating benzene, toluene, styrene, PhA, biphenyl, phenol, nitrobenzene, and DEB. We see that our reaction temperature is the lowest to achieve complete hydrogenation. As stated above, solid organic hydrogen getters are usually employed to eliminate the hydrogen in confined spaces, ensuring hydrogen safety. Figure 6c and Supplementary Table 8 compare the hydrogen uptake capacity of DEB-

containing composites using Pd and Pd-based multifunctional catalysts. Our PdPtRu-contained nanocatalysts exhibit the highest catalytic activity for DEB than other Pd catalysts[22–24,47,48].

In summary, our HEA catalyst has a universal ability for ambient hydrogenation of diverse aromatic derivatives, offering significant potential applications in petrochemical refining, pharmaceuticals, chemical industries, and hydrogen safety and storage (Fig. 6d). Importantly, hydrogenation at ambient conditions and low pressure requires no specific equipment or additional energy, allowing for minimal energy consumption and cost-effectiveness, offering advantages for fine chemical products and organic synthesis.

## Discussion

A series of PdPtRu-containing multifunctional alloy nanocatalysts are synthesized using a simple solvothermal method. These catalysts can hydrogenate solid C-C unsaturated bonds, including alkynyl and phenyl groups, simultaneously and completely at 25 °C under ≤1 bar $H_2$, without the need for any solvent or promoter. The 100% reactivity realizes a threefold increase in hydrogen uptake for DEB. Experimental and theoretical results highlight the role of PdPtRu alloying in enabling the hydrogenation of the phenyl group, with all elements exhibiting a synergistic effect in regulating the overall reaction rate. Significantly, this catalyst is also effective for hydrogenating various aromatics at ambient temperature, with most achieving nearly 100% reactivity in a solvent-free manner. This approach offers a new solution for the ambient hydrogenation of aromatics as well as hydrogen elimination

and control. Future efforts will focus on the large-scale preparation of PdPtRu-containing HEA nanocatalysts and exploring the catalytic hydrogenation for different functional groups, such as carbon-oxygen and carbon-nitrogen groups.

# Methods

## Materials

Carbon nanofiber (CNF) was bought from Nanjing Xianfeng Nano Material Technology Company. Palladium (II) acetylacetonate (Pd(acac)$_2$, 99%), platinum (II) acetylacetonate (Pt(acac)$_2$, 97%), nickel (II) acetylacetonate (Ni(acac)$_2$, 95%), ruthenium (III) acetylacetonate (Ru(acac)$_3$, 97%), copper (II) acetylacetonate (Cu(acac)$_2$, 97%), styrene (99%), phenylacetylene (PhA, 97%), diphenylacetylene (99%) and biphenyl (99.5%) were supplied by Aladdin. Nitric acid (HNO$_3$, 65 - 68%), ethanol (99.7%), acetone (99.5%), and toluene (99.5%) were purchased from Chengdu Colon Chemical Company. 1,4-bis(phenylethynyl)benzene (DEB, 97%) was bought from Alfa Aesar.

## The oxidation of CNFs by HNO$_3$

The CNF (1 g) were added to 100 mL HNO$_3$ and were heated to reflow at 80 °C for 6 h. Then, the mixture was cooled to room temperature, and the solid was collected by centrifuging at 8000 rpm for 10 min. After the product was washed several times with deionized water and freeze-dried for 48 h, the modified carbon nanofibers, containing more carboxylic groups, were obtained and named CNFs.

## Preparation of the high-entropy alloy nanocatalyst

The metal precursor solutions, with a single metal concentration of $1 \times 10^{-3}$ mol/L, were prepared by dissolving the acetylacetone metal salts (Pd(acac)$_2$, Pt(acac)$_2$, Ru(acac)$_3$, Ni(acac)$_2$, and Cu(acac)$_2$) into ethanol and acetone mixed solution (volume ratio of 1:1). Then, 2 mg CNFs and 7 mL of the above acetylacetone metal precursor solution was stirred for 8 h under ultrasonic to form the uniform CNFs/metal precursor mixture. Next, it was transferred into a Teflon-lined steel autoclave, sealed, and heated at 220 °C for 4 h to prepare the nanocatalyst. After that, the autoclave was cooled to ambient temperature in the furnace, and the solid product was collected by centrifuging at 5000 rpm for 10 min. Finally, the high-entropy alloy nanocatalyst, PdPtRuCuNi/CNFs, was obtained by washing twice with ethanol and acetone and drying at 60 °C for 6 h.

## Preparation of others nanocatalysts

The quaternary alloy catalysts (PtRuCuNi/CNFs, PdRuCuNi/CNFs, PdPtCuNi/CNFs, PdPtRuNi/CNFs, and PdPtRuCu/CNFs), trinary alloy catalyst (PdPtRu/CNFs) and individual metal catalysts (Pd/CNFs, Pt/CNFs, Ru/CNFs,) were synthesized using the similar method with the total metal concentration was $5 \times 10^{-3}$ mol/L.

## Preparation of catalysts/DEB composites

The catalysts/DEB composites were prepared by thoroughly grinding the catalysts and DEB powders with a mass ratio of 1:3 to reach sufficient inter-contact.

## Materials characterization

The phase structure of PdPtRuCuNi/CNFs was measured by an X-ray diffractometer (XRD). TEM micrographs and elemental mapping were obtained with a field-emission transmission electron microscopy (FETEM) equipped with an energy-dispersive X-ray (EDX) detector operated at 200 kV. The high-resolution transmission electron microscopy (HRTEM) images were observed by high-angle annular dark field scanning transmission electron microscopy (HAAF-STEM). The atomic ratio was analyzed by inductively coupled plasma optical emission spectroscopy (ICP-OES). Before measurement, the nanocatalysts were dissolved in HNO$_3$ by microwave digestion. X-ray photoelectron spectroscopy (XPS) was used to analyze the valence states of the

PdPtRuCuNi HEA catalyst, quaternary alloy catalysts, and monometal catalysts using a monochromatic Al Kα source (15 mA, 14 kV).

## X-ray absorption measurement and analysis

X-ray absorption fine structure (XAFS) spectroscopy was carried out using the RapidXAFS HE Ultra by transmission mode at 20 kV and 20 mA, and the Si (551) spherically bent crystal analyzer with a radius of curvature of 500 mm was used for Ni; the Si (553) spherically bent crystal analyzer with a radius of curvature of 500 mm was used for Cu; the Si (771) spherically bent crystal analyzer with a radius of curvature of 500 mm was used for Pt. The obtained XAFS data was processed in Athena (version 0.9.26) for background, pre-edge line and postedge line calibrations. Then Fourier transformed fitting was carried out in Artemis (version 0.9.26). The k$^3$ weighting, k-range of 2- 9 Å$^{-1}$ and R range of 1.2 −3 Å were used for the fitting. The four parameters, coordination number, bond length, Debye-Waller factor and E$_0$ shift (CN, R, σ$^2$, ΔE$_0$) were fitted without anyone was fixed, constrained, or correlated.

## H$_2$ temperature programmed desorption

H$_2$ temperature programmed desorption (H$_2$-TPD) was conducted using a homemade system equipped with a quadrupole mass spectrometer (QMS), a quartz tubular flow microreactor, and a programmable temperature controller. The quadrupole mass spectrometer was coupled with the microreactor with stainless steel capillary to ensure online analysis. A K-type thermocouple was placed within the catalyst bed to record the real-time temperature. Typically, these as-synthesized PdPtRuCuNi/CNFs and the commercial Pd/C samples (-50 mg) were treated at 200 °C for 30 min under a flow of Ar (50 mL·min$^{-1}$) to remove the absorption water and then cooled down to room temperature. Next, the flowing Ar was purged into the samples until a stable H$_2$ baseline was obtained. H$_2$-TPD experiments were conducted by heating catalysts to 200 °C at a constant ramp rate in a flow of Ar (50 mL·min$^{-1}$) to trace H$_2$ (m/z = 2) signals with QMS.

## Hydrogenation measurements

The hydrogenation of catalysts/DEB composites was tested using an automatic hydrogen uptake tester at 25 °C under 0-1 bar H$_2$. Before hydrogen uptake measurement, the specimen was pre-baked in a vacuum at 50 °C for at least 10 h to remove any water vapor previously adsorbed. Twenty different equilibrium pressures at 0-1 bar (0.05 bar, 0.1 bar, 0.15 bar, 0.2 bar, up to 1 bar) were set in the hydrogen uptake measurement from the initial to the final stages, and the sample was allowed to uptake H$_2$ until the hydrogen pressure dropped to the pre-set value. The morphology and elemental mapping of catalysts/DEB composites before and after hydrogenation were measured by a field-emission scanning electron microscope (FESEM) with an EDX detector at 10 kV. The chemical composition of DEB before and after hydrogenation was characterized by nuclear magnetic resonance spectroscopy (NMR). The hydrogenated products were analyzed from the NMR spectra, and the conversion was calculated according to the Eq. 1. the $A_{uh}$ is the area of hydrogenated carbon-carbon unsaturated bond, the $A_{uo}$ is the area of original carbon-carbon unsaturated bonds. The yield of hydrogenated products was calculated by the Eq. 2. The $A_{uhs}$ is the Area of hydrogenated carbon-carbon unsaturated bond in the single product, the $A_{uho}$ is the area of carbon-carbon unsaturated bonds of overall hydrogenated products.

$$Conversion = \frac{A_{uh}}{A_{uh} + A_{uo}} \times 100\% \qquad (1)$$

$$Yield = \frac{A_{uhs}}{A_{uho}} \times 100\% \qquad (2)$$

The products in the solvent were characterized by a Gas Chromatography-Mass Spectrometer (GC-MS).

## Calculation method

DFT calculations were performed using the Vienna Ab Initio Simulation Package[49] with the projector augmented wave (PAW)[50,51] method. The electronic exchange and correlation interactions were described by the Perdew–Burke–Ernzerhof (PBE)[52] form of generalized gradient approximation (GGA). The electronic wave function was expanded in plane wave basis sets with a cutoff energy of 500 eV, and the electronic self-consistent convergence was set to $10^{-5}$ eV. For the integration over the Brillouin zone, the Monkhorst–Pack[53] $2 \times 3 \times 1$ $k$-point grid was adopted for all the surface calculations. The spin polarization was included for Cu and Ni atoms. The dispersion correction (PBE-D3) with Becke–Jonson damping was added to the surface and used for adsorption calculations[54,55].

The close-packed (111) surfaces of the pure metals Pd, Pt, Ru, Cu, and Ni, as well as the HEA, $Pd_{21}Pt_{13}Ru_{20}Cu_{24}Ni_{22}$, were built to explore the adsorption of PhA and $H_2$. All the (111) surface slabs were constructed using $4 \times 4$ unit cells, comprising four atom layers and summing up to 64 atoms. These slabs were reoriented to an orthogonal basis. A vacuum layer of 15 Å was introduced in the $z$ direction to prevent interactions between periodic images. For structural optimization, the two bottom layers were fixed to mimic bulk structures, while the top two layers (and the adsorbates) were permitted to relax fully. For the single metals, the DFT-optimized lattice constant for each metal's face-centered cubic (fcc) structure was employed to build the slabs, namely 3.885, 3.924, 3.778, 3.630, and 3.519 Å for Pd, Pt, Ru, Cu, and Ni, respectively. For the HEA, as its optimal lattice constant could vary slightly with different element occupations, we developed five special quasi-random structures (SQSs)[56] each with 108 atoms and fitted an Equation of States (EOS) curve for each SQS, according to the Birch-Murnaghan 4th-order EOS. The average of the optimal lattice constants from these five SQSs, calculated to be 3.771 Å, was adopted as the initial lattice constant when building the slabs. We constructed six individual slabs with random elemental occupations (Supplementary Fig. 23) to explore the vast variability of adsorption sites.

For the adsorption of PhA, the initial adsorption configuration situated the phenyl group above a 3-fold hollow site, with the acetylenic group forming di-μ bonding. A visual illustration of this configuration is provided in Supplementary Fig. 24. In the context of HEA, to reveal the site-specific adsorption energies ($\Delta E_{ads-PhA}$) on its surfaces, we computed the $\Delta E_{ads-PhA}$ for each possible adsorption site present on the HEA surface. As a visual guide, Supplementary Fig. 24 illustrates the 32 considered adsorption sites on each slab, with the hollow sites, where the phenyl group resides, being highlighted by yellow triangles. Thus, a total of 192 adsorption sites, spanning six slabs, were evaluated for PhA adsorption on HEA. For the adsorption of $H_2$, both the hollow and bridge sites on the (111) surfaces were taken into account. To elucidate this, Supplementary Fig. 25a and b-d depict typical adsorption configurations with an $H_2$ molecule above a hollow site and above the bridge sites, respectively. On our constructed (111) surfaces, there are 32 hollow sites, identical to those illustrated for PhA in Supplementary Fig. 24, and 48 bridge sites. Consequently, 192 hollow sites and 288 bridge sites, summing up to 480 sites, were considered for $H_2$ adsorption.

The adsorption energy ($\Delta E_{ads}$) was defined as,

$$\triangle E_{ads} = E_{adsorbate/surface} - E_{adsorbate} - E_{surface} \qquad (3)$$

where $E_{adsorbate/surface}$, $E_{adsorbate}$, and $E_{surface}$ represent the energies of the system with adsorbate attached to the surface, the isolated adsorbate, and the unoccupied surface, respectively.

To determine minimum energy paths and saddle points on the potential energy surface of the ethylbenzene hydrogenation reaction, the Nudged Elastic Band (NEB)[42] and Dimer methods[57] were applied. The NEB method was used to determine an approximate saddle point geometry on the minimum energy path between the known reactants and products. Based on the saddle point identified by the NEB method, the DIMER method was performed to determine an exact geometry of the transition state, also with all forces below 0.03 eV/Å to allow a subsequent vibrational analysis.

The energy barrier for the hydrogenation was defined as,

$$\triangle E_a = E(TS) - E(IS) \qquad (4)$$

where $E(TS)$ and $E(IS)$ represent the energies of the hydrogenation transition state (TS) and the initial state (IS), respectively.

## Data availability

The data supporting the findings of this study are available within the article and Supplementary Information files, and also are available from the corresponding author on request. Source data are provided with this paper.

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

## Acknowledgements

This work was financially supported by the National Key R&D Program of China (Grant No: 2021YFA1202300), the Joint Fund of the National Natural Science Foundation of China and the China Academy of Engineering Physics (Grant No: U2030207), National Natural Science of Foundation of China (NSFC) (Grant Nos: 52101255, 52371223), and the SPC-Lab Research Fund (WDZC202001).

## Author contributions

K.G., and M.B.S. conceived and supervised the research. Z.K.J., Z.D.L., and M.Z.L. designed the experiments. Z.K.J., and G.Z.Y. performed data analysis of NMR. Q.W., X.R.Y., and D.G.X. performed the DFT simulations. C.Y.Z., and Y.X.Y. performed the TPD

experiment. R.X.Q. performed and analysed the EXAFS. Y.K.G., and H.W.L. drew the schematic diagram, and Z.K.J. wrote the paper. Q.W., Y.X.Y., and Y.G.Y. revised the paper. All authors discussed the results and commented on the manuscript.

## Competing interests

The authors declare that they have no competing financial interests or personal relationships that could have appeared to influence the work reported in this paper.
