## [Peer Review File · Nature Communications]

Ambient hydrogenation of solid aromatics enabled by a high entropy alloy nanocatalystREVIEWER COMMENTS

Reviewer #1 (Remarks to the Author):

Ambient hydrogenation of solid aromatics enabled by a high entropy alloy nanocatalyst by Jing et al. deals with the hydrogenation in the solid phase of 1,4-bis(phenylethynyl)benzene (DEB) at 25 °C under ≤ 1 bar H₂, using a multimetallic catalyst. I do think that the work is well thought, executed and presented, and it might be of interest for publication in this journal. I do think also that some points need to be clarified before.

I am not convinced that a Pd/C is the best choice to establish a comparison. Pd based catalysts are not good in arene hydrogenation, as the author's findings demonstrate that clearly. A Ru based catalysts, such as Ru/C, should be employed, as in my opinion Ru based catalysts are the state-of-the-art catalysts to challenge. Thus, authors should do some extra experiments using commercial Ru/C, being careful to maintain same metal charge.

The catalyst is poorly characterised. We cannot deduce that all metals are in the same nanoparticle by any means. DRX signals are too broad; XPS does not show any interaction, TEM mapping is not magnified enough, the planes distance established by HRTEM do not tell much either. I do propose to characterize the catalysts by exafs and HRTEM/EDX. For the later, several EDX should be measured in individual nanoparticles, small and large, as the population is broad.

There is no characterization of the other synthesized catalysts containing four metals; DRX, XPS and HRTEM of these materials should be performed.

PdPtRu-containing catalysts appear to outperform other catalysts. It is surprising, seeing that the synthesis is straightforward, that a trimetallic PdPtRu catalyst was not synthesized and used as catalyst. I definitely recommend doing so, with the characterization that comes with.

Last, I am not sure if we can call solid state, if at the end of the catalysis the solid is not anymore solid, it seems that catalysis at some point is solventless but not solid. Authors should think about that.

The authors should try also to deshydrogenate the molecule using the best catalyst.

As summary, I do think that the results are very appealing, hydrogenation at ambient conditions, but the origin of the activity is not well established and some other elements should be included, as an effort in the characterization of the catalysts as well as a comparison with catalysts of interest.

Reviewer #2 (Remarks to the Author):

This study reported that the prepared PdPtRuCuNi high entropy alloy (HEA) nanocatalyst on acid-treated CNF achieves an exceptional 100% hydrogenation of carbon-carbon unsaturated bonds, including alkynyl and phenyl groups, in solid DEB at 25 °C under ≤ 1 bar H₂. This paper is well organized, and the synergistic catalytic effect was elucidated based on the systematic characterization and mechanistic investigation using DFT calculation. I agree that this work presents a useful advance on understanding of new phenomena and the development of novel HEA catalyst. This study has potential to be published but contains some insufficient points. The following issues should be considered before publication.

1) The comparison of the hydrogen uptake and catalytic activity was performed with the conventional Pd/C catalyst. The authors should employ with Pd/CNF prepared with the same precursor, support and preparation method.

- 2) The total amount of metals for the compared catalysts are unclear. According to Supplementary Table 1, the total amount of metals of quinary-component PdPtRuCuNi/CNF was 120 wt%. The authors compared the hydrogen uptake and catalytic activity with conventional Pd/C catalysts. How much the amount of Pd. It should be unified with quinary-component catalyst.
- 3) In relation to the above comment, the amount of ternary and quaternary system should be clarified.
- 4) The solid-solid hydrogenation is interesting, but the results under liquid-phase reaction using appropriate solvent should be provided.
- 5) This paper lacks kinetic analysis. The effect of hydrogen pressure and amount of catalyst should be provided at least.
- 6) In my opinion, the comparison of the reaction temperature with the previously reported system is insufficient. Of course, the authors summarize the detail reaction conditions in Supplementary Table 5 & 6. But is it possible to provide the activation energy determined by Arrhenius plot

Point-by-Point Response to the Comments

(Blue: reviewer's comments; Black: our response)

Reviewers' comments:

Reviewer #1:

Ambient hydrogenation of solid aromatics enabled by a high entropy alloy nanocatalyst by Jing et al. deals with the hydrogenation in the solid phase of 1,4-bis(phenylethynyl)benzene (DEB) at 25 °C under ≤ 1 bar H_2 , using a multimetallic catalyst. I do think that the work is well thought, executed and presented, and it might be of interest for publication in this journal. I do think also that some points need to be clarified before.

Response: We appreciate the referee's positive comments and insightful suggestions to help us significantly improve the quality of our manuscript. We have performed additional experiments and revised our manuscript carefully, and sincerely hope that our revisions have satisfactorily addressed the reviewer's concerns.

1. I am not convinced that a Pd/C is the best choice to establish a comparison. Pd based catalysts are not good in aromatics hydrogenation, as the author's findings demonstrate that clearly. A Ru based catalysts, such as Ru/C, should be employed, as in my opinion Ru based catalysts are the state-of-the-art catalysts to challenge. Thus, authors should do some extra experiments using commercial Ru/C, being careful to maintain same metal charge.

Response: We appreciate the constructive comments raised by the referee. We have performed the Ru/C catalyzed DEB at 25 °C and ≤ 1 bar H_2 pressure. The results show that the hydrogen uptake capacity of Ru/C-DEB composites is nearly 0 cm^3/g (**Fig. R1**). The 1H and ^{13}C NMR spectra of DEB before and after hydrogenation show no difference, indicating that the chemical structure does not show change, so the conversion of DEB over Ru/C is near 0% (**Fig. R2**). Therefore, although Ru/C is a state-of-the-art catalyst for hydrogenation, it has no catalytic activity for hydrogenation of DEB under room temperature and low H_2 pressure due to the higher reaction energy barrier.

Fig. R1 The hydrogen uptake capacity of different catalysts, including Ru/C. **a** The hydrogen uptake curves and reaction rates with time. **b** Hydrogen uptake capacities, showing Ru/C has negligible reactivity for DEB hydrogenation under our condition.

Fig. R2 **a** ^{13}C NMR, and **b** ^1H spectra of DEB before and after hydrogenation. **c** The conversion of C-C unsaturated bonds (alkynyl and phenyl groups) in the hydrogenated DEB molecules.

Modification:

1) Main text, page 7, line 128-131, added text “In addition, the commercial Pd/C and Ru/C are used as contrast catalysts, the hydrogen uptakes of Pd/C and Ru/C-DEB composites are 186 cm^3/g and 0 cm^3/g (Fig. 3b), respectively, which are significantly lower than PdPtRuCuNi/CNFs, and Ru/C with no catalytic activity for DEB in mild condition.”

2) Main text, **Fig. 3b**, added the histogram of the final hydrogen uptake capacities for commercial Pd/C-DEB, Ru/C-DEB and PdPtRuCuNi/CNFs-DEB.

3) **Supplementary Fig. 7** of the catalytic hydrogenation experiments by different catalysts added to the revised Supplementary Information.

2. The catalyst is poorly characterized. We cannot deduce that all metals are in the same nanoparticle by any means. XRD signals are too broad; XPS does not show any interaction, TEM mapping is not magnified enough, the planes distance established by HRTEM do not tell much either. I do propose to characterize the catalysts by EXAFS and HRTEM/EDX. For the later, several EDX should be measured in individual nanoparticles, small and large, as the population is broad.

Response: Thanks for raising this point. As suggested, we have conducted EDX mapping for individual and groups of nanoparticles, and the results are provided in **Fig. R3**. The Pd, Pt, Ru, Cu, and Ni elements coexist in the nanoparticles, demonstrating the formation of a solid solution alloy involving all five elements.

Fig. R3 The EDS mapping of PdPtRuCuNi/CNFs. **a** Individual nanoparticle. **b** Several nanoparticles.

To further demonstrate the metal interaction of PdPtRuCuNi HEA, we compared the HEA with the monometallic catalysts (Pd/CNFs, Pt/CNFs, Ru/CNFs, Cu/CNFs, and Ni/CNFs). They were synthesized using the same method and characterized by XPS (X-ray photoelectron spectroscopy), the results are shown in **Fig. R4**. Compared with mono-metals, Ni shift to higher binding energy (about 0.1 eV or 0.3 eV), Pd, Pt, Ru, and Cu shift to lower binding energy (0.6 eV~1.2 eV) in the PdPtRuCuNi HEA, which means noticeable electron transfer among each other, suggesting successful alloying rather than separating.

Fig. R4 The XPS spectra of PdPtRuCuNi HEA and monometal catalyst. **a** Pd 3d, **b** Pt 4f, **c** Ru 3p, **d** Cu 2p, and **e** Ni 2p XPS high-resolution spectra.

The Pd and Ru elements need to perform at higher energy (Pd 24300 eV-24500 eV and Pt 24300-24500 eV) and in a different mode. In China, only the Shanghai Synchrotron Radiation Facility can carry out this work, but they start this mode in the second half of the year, and the specific time is uncertain, so only the Pt, Ni and Cu are tested. The X-ray absorption near edge structure (XANES) spectra were carried out using the RapidXAFs HE Ultra by transmission mode at 20 kV and 20 mA, and results are shown in **Fig. R5 a-c**. The Pt, Cu, and Ni absorption energies for PdPtRuCuNi/CNFs are similar to those of metal foil, indicating the dominance of metallic-state elements. The last half is slightly dissimilar in shape and intensity, meaning the interaction among each element. The Fourier transforms of these EXAFS are displayed in **Fig. R5 d-f**. The distances of Me–Me in PdPtRuCuNi HEA are shorter than that of metal foils and the blue line intensity of PdPtRuCuNi HEA is much lower than that of black metal foil, revealing the bond lengths and coordination numbers of Pt, Cu, and Ni are different from the

high-entropy catalysts, which further confirm by the EXAFS fitting, and the bond length (R) and coordination numbers are surmised in **Table R1**. These results indicate the alloying of these metals and strong electronic interactions in the PdPtRuCuNi HEA. Above these analyses, we demonstrate that all metals are distributed uniformly in the same nanoparticle with strong electronic interactions, thus proving alloying in the PdPtRuCuNi HEA.

Fig. R5 The characterization of X-ray near-edge absorption. **a-c** The X-ray absorption near edge structure (XANES) spectra of PdPtRuCuNi/CNFs and monometal catalysts. **a** The Pt L3-edge. **b** The Cu K-edge. **c** The Ni K-edge. **d-f** Fourier transform EXAFS spectra of PdPtRuCuNi/CNFs and monometal catalysts. **d** Pt. **e** Cu. **f** Ni.

Table. R1 EXAFS fitting parameters at the Cu and Ni K-edge, Pt L3-edge ($S_0^2=0.90^*$)

Sample	Path	C.N.	R (Å)	$\sigma^2 \times 10^3$ (Å ²)	ΔE (eV)	R factor
Cu foil	Cu-Cu	12*	2.54±0.01	10.6±2.2	6.7±0.4	0.002
Cu	Cu-Cu	6.2±5.8	2.55±0.04	10.9±12.2	1.9±1.9	0.020
	Cu-Pd	2.0±0.7	2.56±0.06	5.0±5.9		
Ni foil	Ni-Ni	12*	2.48±0.01	6.2±0.2	7.4±0.3	0.001
Ni	Ni-Ni	3.3±0.7	2.45±0.01	6.8±1.7	-8.8±1.1	0.010
	Ni-Pd	1.7±1.0	2.64±0.05	17.7±9.5	11.8±1.5	
Pt foil	Pt-Pt	12*	2.77±0.01	4.8±0.2	3.6±0.5	0.001
Pt	Pt-Cu	6.4±2.7	2.60±0.02	16.1±5.5	4.0±1.7	0.004
	Pt-Pd	4.8±1.1	2.71±0.03	14.0±4.2		

$C.N.$: coordination numbers; R : bond distance; σ^2 : Debye-Waller factors; ΔE : the inner potential correction. R factor: goodness of fit. * fitting with fixed parameter.

Modification:

1) Main text, page 5-6, line 99-111, added text “Compared to their individual metals, the binding energies of Pd 3d, Pt 4f, Ru 3p, and Cu 2p shift to the lower energies (Fig. 2f-i), while Ni2p exhibits a slight shift to higher energies (Fig. 2g). This suggest that alloying these metals result in a strong electronic interaction within the PdPtRuCuNi HEA, which could change the surface electronic structure and potentially enhance its catalytic performance⁴⁰. This result has also been demonstrated by X-ray absorption near edge structure (XANES) spectra (Supplementary Fig. 4a-c).....”.

2) Main text, **Fig. 2f-g**, added the XPS high-resolution spectra for PdPtRuCuNi HEA and monometallic catalyst.

3) **Supplementary Fig. 4** and **Supplementary Table 4** about the EXAFS has been added to the revised Supplementary Information.

3. There is no characterization of the other synthesized catalysts containing four metals; XRD, XPS and HRTEM of these materials should be performed.

Response: Thanks for the suggestion. The five quaternary alloy catalysts (PtRuCuNi, PdRuCuNi, PdPtCuNi, PdPtRuNi, PdPtRuCu) are characterized by XRD, XPS, and HRTEM; the results are shown in **Fig. R6~Fig. R16**. Briefly, the five quaternary alloys are the single-phase alloy with a fcc structure except for PtRuCuNi; the metal elements distribute uniformly on the nanoparticles, the metals exist mainly in their zero-valence states in the alloy except for Ni. To our surprise, Pd is the crucial factor in forming single-phase alloys.

Fig. R6 The XRD patterns of quaternary alloy catalysts.

Fig. R7 The TEM analyses of PtRuCuNi/CNFs. **a** TEM morphology. **b** HRTEM image. **c** selective electron diffraction pattern. **d** EDS mapping.

Fig. R8 The TEM analyses of PdRuCuNi/CNFs. **a** TEM morphology, **b** HRTEM image, **c** selective electron diffraction pattern, **d** the EDS mapping.

Fig. R9 The TEM analyses of PdPtCuNi/CNFs. **a** TEM morphology, **b** HRTEM image, **c** selective electron diffraction pattern, **d** the EDS mapping.

Fig. R10 The TEM analyses of PdPtRuNi/CNFs. **a** TEM morphology, **b** HRTEM image, **c** selective electron diffraction pattern, **d** the EDS mapping.

Fig. R11 The TEM analyses of PdPtRuCu/CNFs. **a** TEM morphology, **b** HRTEM image, **c** selective electron diffraction pattern, **d** the EDS mapping.

Fig. R12 The XPS results of PtRuCuNi/CNFs. **a** survey spectrum. **b** Pt 4f, **c** Ru 3p, **d** Cu 2p, and **e** Ni 2p high-resolution spectra.

Fig. R13 The XPS results of PdRuCuNi/CNFs. **a** survey spectrum. **b** Pd 3d, **c** Ru 3p, **d** Cu 2p, and **e** Ni 2p high-resolution spectra.

Fig. R14 The XPS results of PdPtCuNi/CNFs. **a** survey spectrum. **b** Pd 3d, **c** Pt 4f, **d** Cu 2p, and **e** Ni 2p high-resolution spectra.

Fig. R15 The XPS results of PdPtRuNi/CNFs. **a** survey spectrum. **b** Pd 3d, **c** Pt 4f, **d** Ru 3p, and **e** Ni 2p high-resolution spectra.

Fig. R16 The XPS results of PdPtRuCu/CNFs. **a** survey spectrum. **b** Pd 3d, **c** Pt 4f, **d** Ru 3p, and **e** Cu 2p high-resolution spectra.

Modification:

1) Main text, page 9, line 173-176, added text and “their phase structures, chemical compositions, and morphologies have been characterized in detail (Supplementary Figs. 9-19).”.

2) **Supplementary Fig. 9-19** about the characterization of quaternary alloy catalyst has been added to the revised Supplementary Information.

4. PdPtRu-containing catalysts appear to outperform other catalysts. It is surprising, seeing that the synthesis is straightforward, that a trimetallic PdPtRu catalyst was not synthesized and used as catalyst. I definitely recommend doing so, with the characterization that comes with.

Response: We appreciate the constructive comments raised by the referee. The PdPtRu/CNFs catalyst was synthesized and characterized by XRD; the results show that PdPtRu/CNFs are a single-phase alloy with a fcc structure (**Fig. R17**). Then, we carried out the hydrogenation experiment at the same conditions (25 °C and ≤ 1 bar). The product was further measured by NMR and was found to be the same as that of PdPtRuCuNi/CNFs. This suggests that both PdPtRu/CNFs and PdPtRuCuNi/CNFs have excellent catalytic activity and further proves PdPtRu to be the leading active site in DEB hydrogenation.

Fig. R17 The XRD patterns of PdPtRu/CNFs catalyst.

Fig. R18 a ^{13}C and b ^1H NMR spectra of DEB before and after hydrogenation by PdPtRu/CNFs. c the conversion.

Modification:

1) Main text, page 9, line 186-187, added text “These findings have also been demonstrated by PdPtRu/CNFs catalyst (Supplementary Fig. 21 and 22)”.

2) The synthesis of quaternary alloy, PdPtRu and monometallic catalysts has been updated in the revised article. **Supplementary Fig. 21** and **22** about the XRD and catalytic hydrogenation for PdPtRu/CNFs has been added to the revised Supplementary Information.

5. Last, I am not sure if we can call solid state, if at the end of the catalysis the solid is not anymore solid, it seems that catalysis at some point is solventless but not solid. Authors should think about that.

Response: We appreciate the insightful comments from the referee. We strongly agree with the solvent-free status proposed by the reviewer. Here the solid state refers to the starting reactant as the solid state.

Modification: We have modified the solid-state into solventless and identified it in the red letters, please see the revised article.

6. The authors should try also to dehydrogenate the molecule using the best catalyst.

Response: Thanks for the suggestion. The hydrogen absorption and desorption experiment of PdPtRuCuNi/CNFs-DEB has been performed at 25 °C, 50 °C, and 100 °C under ≤ 1 bar H_2 . In the desorption stage (1 bar~0.5 bar), the amount of hydrogen uptake does not change (**Fig. R18**), indicating that the hydrogenation process is irreversible, which is very suitable for an organic hydrogen getter.

Fig. R19 The hydrogen absorption and desorption curves of PdPtRuCuNi/CNFs-DEB at various temperature.

Modification:

Supplementary Fig. 8a about absorption and desorption curves with pressure at different temperatures has been added to the revised Supplementary Information.

Reviewer #2:

This study reported that the prepared PdPtRuCuNi high entropy alloy (HEA) nanocatalyst on acid-treated CNF achieves an exceptional 100% hydrogenation of carbon-carbon unsaturated bonds, including alkynyl and phenyl groups, in solid DEB at 25 °C under ≤ 1 bar H_2 . This paper is well organized, and the synergistic catalytic effect was elucidated based on the systematic characterization and mechanistic investigation using DFT calculation. I agree that this work presents a useful advance on understanding of new phenomena and the development of novel HEA catalyst. This study has potential to be published but contain some insufficient point. The following issue should be considered before publication.

Response: We appreciate the referee's positive comments and valuable suggestions for improve our work. We have carefully addressed all suggestions and performed additional experiments. A new version is attached for further review.

1. The comparison of the hydrogen uptake and catalytic activity was performed with the conventional Pd/C catalyst. The authors should employ with Pd/CNF prepared with the same precursor, support and preparation method.

Response: We appreciate the constructive comments raised by the referee. The Pd/CNFs catalyst with the 60.3 wt% load was prepared using the same precursor, support, and preparation method. We evaluated the catalytic activity at 25 °C under ≤ 1 bar; the results show that the hydrogen uptake and conversion of unsaturated C-C bond of Pd/CNFs-DEB are 198 cm^3/g and 26.5%, respectively, which have similar catalytic activity as Pd/C (186 cm^3/g and 25.8%) (**Fig. R20**), illustrating that the carbon support and loading amount does not influence on the catalytic hydrogenation property of Pd catalyst.

Fig. R20 The hydrogenation performance of DEB catalyzed by PdPtRuCuNi/CNFs and Pd/CNFs at 25 °C under ≤ 1 bar H_2 . **a** The final hydrogen uptake capacities. **b** The curves of hydrogen uptake and reaction time. **c** ^{13}C NMR spectra of DEB before and after hydrogenation. **d** The conversion of C-C unsaturated bonds (alkynyl and phenyl groups) in the hydrogenated DEB molecules.

Modification:

- 1) We have used the Pd/CNFs to replace the Pd/C as the compared catalyst and identified it in the red letters; please see the revised article.
- 2) **Fig. 3a** and **d-f** about the catalytic hydrogenation for PdPtRuCuNi /CNFs and Pd/CNFs has been updated to the revised article.

2. The total amount of metals for the compared catalysts are unclear. According to Supplementary Table 1, the total amount of metals of quinary-component PdPtRuCuNi/CNF was 120 wt%. The authors compared the hydrogen uptake and catalytic activity with conventional Pd/C catalysts. How much the amount of Pd. It should be unified with quinary-component catalyst.

3. In relation to the above comment, the amount of ternary and quaternary system should be clarified.

Response:As suggested by the two comments of the reviewer, we have summarized the loading and element distribution in all our samples (**Table R2**). The catalysts were measured by ICP-OES, and the total amount and the amount of monometal were calculated. The total amount of metals in quinary-component PdPtRuCuNi/CNFs was 55.2 wt%. Pd in PdPtRuCuNi/CNFs and Pd/C are 12.5 wt% and 4.9 wt%.

Table R2 The total load and mono-metal load of catalysts.

samples	Pd (wt%)	Pt (wt%)	Ru (wt%)	Cu (wt%)	Ni (wt%)	Toal (wt%)
PdPtRuCuNi/CNFs	12.5	15.0	12.1	8.3	7.3	55.2
PtRuCuNi/CNFs		17.8	11.3	9.3	6.9	45.3
PdRuCuNi/CNFs	16.3		12.0	10.2	9.0	47.5
PdPtCuNi/CNFs	14.9	16.2		10.9	7.4	49.4
PdPtRuNi/CNFs	16.2	13.5	12.9		8.2	50.8
PdPtRuCu/CNFs	9.8	19.2	7.2	10.4		46.6
PdPtRu/CNFs	19.5	20.2	13.9			53.6
Ru/C			4.6			4.6
Pd/C	4.9					4.9
Pd/CNFs	60.3					60.3

Modification:

- 1) Main text, page 9, line 172, added text “with similar loadings (supplementary table 5)”.
- 2) **Supplementary Table 5** about the loadings for catalysts has been updated in the revised Supplementary Information.

4. The solid-solid hydrogenation is interesting, but the results under liquid-phase reaction using appropriate solvent should be provided.

Response: We appreciate the constructive comments raised by the referee. Here, using diphenylacetylene as the substrate and isopropyl as the solvent, we performed the liquid-phase experiment at 25 °C and 10 bar H₂ pressure. The product was analyzed by gas chromatography-mass spectrometer (GC-MS), demonstrating that the diphenylacetylene could realize 100% conversion (**Fig. R21**), which agrees well with solventless hydrogenation.

Fig. R21 The hydrogenation of diphenylacetylene catalyzed by PdPtRuCuNi/CNFs under liquid-phase. **a** GC-MS results. **b** the conversion.

Modification:

Supplementary Fig. 38 about the hydrogenation in the solvent manner has been updated in the revised Supplementary Information.

5. This paper lacks kinetic analysis. The effect of hydrogen pressure and amount of catalyst should be provided at least.

Response: We appreciate the constructive comments raised by the referee. Here, using diphenylacetylene as the substrate, we performed the hydrogenation experiment at different hydrogen pressures (10 bar, 20 bar, and 30 bar) and amounts of catalyst (catalyst/substrate = 1/3, 1/4, and 1/5). The products were analyzed by nuclear magnetic resonance spectrometer, demonstrating that the diphenylacetylene could realize 100% conversion at different hydrogen pressures (**Fig. R22a, b, and e**). However, the conversion significantly decreases with the catalyst reduction, from 100.0% to 59.7% (**Fig. R22 c, d, and f**). This may be caused by the increasing distance between the atomic hydrogen and the unsaturated carbon-carbon bonds and blocking hydrogen diffusion and migration.

Fig. R20 The diphenylacetylene were catalyzed by PdPtRuCuNi/CNFs at different hydrogen pressures (10 bar, 20 bar, and 30 bar) and amounts of catalyst (1:3,1:4, and 1:5). **a** ^{13}C and **b** ^1H NMR spectra of diphenylacetylene before and after hydrogenation. **e** The conversion at different hydrogen pressure. **c** ^{13}C and **d** ^1H NMR spectra of diphenylacetylene before and after hydrogenation. **f** The conversion at different amount of catalyst.

Modification:

Supplementary Fig. 37 about the diphenylacetylene hydrogenation at various temperatures and amounts of catalyst has been updated in the revised Supplementary Information.

6. In my opinion, the comparison of the reaction temperature with the previously reported system is insufficient. Of course, the authors summarize the detailed reaction conditions in Supplementary Table 5 & 6. But is it possible to provide the activation energy determined by Arrhenius plot?

Response: We appreciate the constructive comments raised by the referee. Determining activation energy by the Arrhenius plot is a good choice. However, we are sorry to say that the hydrogenation reaction of PdPtRuCuNi/CNFs-DEB follows different reaction kinetics at various temperatures (**Fig. R22**). Therefore, it cannot provide the activation energy data calculated by the Arrhenius plot. The catalytic hydrogenation of the benzene ring reaction is complicated, and the activation energies provided hardly by the experiments in the most reported literature, such as supplementary references 6 (Jiang, W. et al. *Appl. Catal. B: Environ.*), 14 (Wu, D. et al. *Nat. Catal.*), 15(Shi, S. et al. *Nat. Commun.*).

Fig. R22 The hydrogen uptake curves and reaction rates with time of PdPtRuCuNi/CNFs-DEB at different temperatures.

Modification:

1) Main text, page 7, line 133-142, added text “Firstly, the hydrogen uptake curves of PdPtRuCuNi/CNFs-DEB were evaluated at various temperatures (25, 50, and 100 °C, **Fig. 3c** and **Supplementary Fig. 8**). The results show that the similar hydrogen uptake values are obtained at 25 and 50 °C, while some decrease of hydrogen uptake at 100 °C because increasing the temperature promote the chemical reaction equilibrium moving in the opposite direction due to the exothermic hydrogenation reaction.....”.

2) **Fig. 3c** about the catalytic hydrogenation at different temperatures has been added to the revised article.

REVIEWERS' COMMENTS

Reviewer #2 (Remarks to the Author):

The authors have carried out further studies on the system and clarified the original questions. The manuscript reads well and now this paper is recommended to publish as revised version.